# PyUUL provides an interface between biological structures and deep learning algorithms

Gabriele Orlando[1,2], Daniele Raimondi[3], Ramon Duran-Romaña[1,2], Yves Moreau[3], Joost Schymkowitz [1,2✉] & Frederic Rousseau [1,2✉]

Structural bioinformatics suffers from the lack of interfaces connecting biological structures and machine learning methods, making the application of modern neural network architectures impractical. This negatively affects the development of structure-based bioinformatics methods, causing a bottleneck in biological research. Here we present PyUUL (https://pyuul.readthedocs.io/), a library to translate biological structures into 3D tensors, allowing an out-of-the-box application of state-of-the-art deep learning algorithms. The library converts biological macromolecules to data structures typical of computer vision, such as voxels and point clouds, for which extensive machine learning research has been performed. Moreover, PyUUL allows an out-of-the box GPU and sparse calculation. Finally, we demonstrate how PyUUL can be used by researchers to address some typical bioinformatics problems, such as structure recognition and docking.

[1] Switch Laboratory, VIB-KU Leuven Center for Brain and Disease Research, Herestraat 49, 3000 Leuven, Belgium. [2] Switch Laboratory, Department of Cellular and Molecular Medicine, KU Leuven, Herestraat 49, 3000 Leuven, Belgium. [3] ESAT-STADIUS, KU Leuven 3000, Belgium. ✉email: joost.schymkowitz@kuleuven.be; frederic.rousseau@kuleuven.be

Structural biology and bioinformatics often lag behind when it comes to the application of cutting-edge Machine Learning (ML) algorithms, highlighting a technological gap between algorithms used at the forefront of ML research and those adopted in other disciplines. The stunning results recently obtained by AlphaFold[1] and RoseTTA fold[2] in protein folding prediction demonstrated how advanced ML, and more specifically deep learning, could be the key to breakthroughs in unsolved biological and biophysical problems. The sometimes significant difference in ML technology between fields of science could indeed open opportunities for technology arbitrage, where methods that are well-known in a specific context, such as computer vision or natural language processing, could be ground-breaking when applied in other context in which they are unknown but potentially valuable[3–5].

In particular, one of the aspects of structural biology that is most severely affected by this uneven technology availability is ML applied to protein structures. Most of the existing structure-based computational approaches only indirectly use the structural information available (i.e., PDB structures[6]). Instead, these methods compute features from PDB structures, such as statistical potentials[7], and then they feed these precomputed values into conventional ML methods, such as Support Vector Machines, Random Forests, or Multilayer Perceptrons[8,9]. This can be a suboptimal solution, since nontrivial structural information might not be directly picked up by statistical potentials and thus be inadvertently discarded in the process. Nonetheless, researchers had to adopt this approach because of (1) the lack of tools to seamlessly translate the complex data structures used to represent 3-dimensional macromolecules structures into formats suitable for easy ML analysis and (2) the lack of flexibility of conventional ML methods when it comes to deal with complex structured input data instead of classical one-dimensional feature vectors.

Early applications of computer vision deep learning techniques to structural biology have already shown their effectiveness for a large number of applications, including protein-binding site prediction[10–12], mutation effect prediction[13], and docking[14]. A broader access of the scientific community to these algorithms will speed up this significant step forward in the field.

In recent years, following breakthroughs in deep learning in fields such as computer vision, image recognition, and natural language processing[15,16], companies such as Google and Facebook have contributed consistently to the democratization of the access to neural network (NN) technologies by developing flexible open-source libraries, such as TensorFlow and Pytorch[17,18]. These libraries let researchers build ad hoc models for any given problem by allowing the use of arbitrarily complex multi-dimensional tensorial structures as input features, thus removing the methodological limitations of classical ML methods mentioned. Since the tools provided by these flexible NN libraries are nowadays accessible to every researcher, the gap that remains to be filled to enable the application of deep learning algorithms in structural biology (i.e., end-to-end learning on biological structures) consists in providing reliable and transparent instruments to seamlessly translate the raw 3D macromolecule structures from PDB into ML-suitable formats. These new formats can then be used directly as input features in the most recent NN architectures, including Transformers and 3D convolutional NN[19].

In this paper, we aim at filling this gap by presenting PyUUL, a Pytorch[18] library designed to process 3D structures of biological molecules (i.e., proteins, drugs and nucleic acids), translating them into differentiable, ML-ready tensorial representations, such as volumetric grids or point clouds. In other words, PyUUL allows researchers to apply any newly developed NN architecture to their favorite structural biology problem by translating the structural data into ML-ready data structures. This translation is completely transparent to the user, therefore masking the intrinsic complexity of the 3D structures. Furthermore, PyUUL follows a completely end-to-end approach, meaning that it can be used as an internal passage of a NN, with backpropagation gradients flowing through it. PyUUL is available as Python package at https://bitbucket.org/grogdrinker/pyuul/.

## Results

**PyUUL supports different structure representations**. PyUUL provides an interface between biological structures, such as PDB structures[6], and deep learning algorithms. The user can choose between three different type of tensor representation of biological structures: voxel-based, surface point cloud, and volumetric point cloud (Fig. 1A).

**Voxel-based representation**. In voxel-based representation, each macromolecule is represented as a 3D box in which every voxel (a 3D pixel) has a size in Angstroms defined by the user (resolution). Every atom is described as a sphere in this space, where the radius corresponds to the atomic radius. Atoms are therefore not represented as point-like entities, but as solid structures. Similarly to pictures' Red, Green, Blue colors, each voxel contains several channels, describing different types of atomic information. In PyUUL each channel specifies the density for a specific atom type. The content and the number of channels is fully customizable by the user. This approach has been successfully used in other studies[10,13,20].

**Surface point-cloud representation**. The second data structure that the user can choose to represent the target 3D biological structures is the surface point-cloud representation. In this setting, the macromolecule is represented with a group of points sampled on its surface. This data representation has recently shown to have several advantages[21,22], and the deep learning algorithms that can efficiently handle it are rapidly gaining popularity[23,24]. While this representation is fairly new in ML, this topic of research is very active in computer vision, in particular in object recognition[25].

This type of representation is often used to address problems in which the information comes from the object surface. While the object is placed in a 3D environment, the information lays on a 2D surface (called manifold). The idea is therefore to efficiently store the information of the surface only, avoiding to use way larger 3D voxel-based volumes. Since this type of representation is often used in face and object recognition[26,27], as well as in automatic car driving[28], many of the algorithms developed in this field are invariant to translations and rotations, making them potentially valuable for protein analysis.

**Volumetric point-cloud representation**. The last representation that is provided by PyUUL is similar to the previous one, but in this case the points are not sampled just on the surface, but also inside the volume of every atom. This can be useful when dealing with large macromolecules that are hard to handle with voxel-based approaches, but where it is important not to lose volumetric information. This method allows sampling points from the volume occupied by every atom.

**PyUUL volumes are deep learning-ready**. A common task in structural biology consists in predicting a specific structural feature of proteins, where this feature depends in a certain way from the 3D structure of the target proteins or of certain regions. This could be the case of real-world tasks, such as protein-protein interaction prediction or active sites/epitopes identification[10,29].

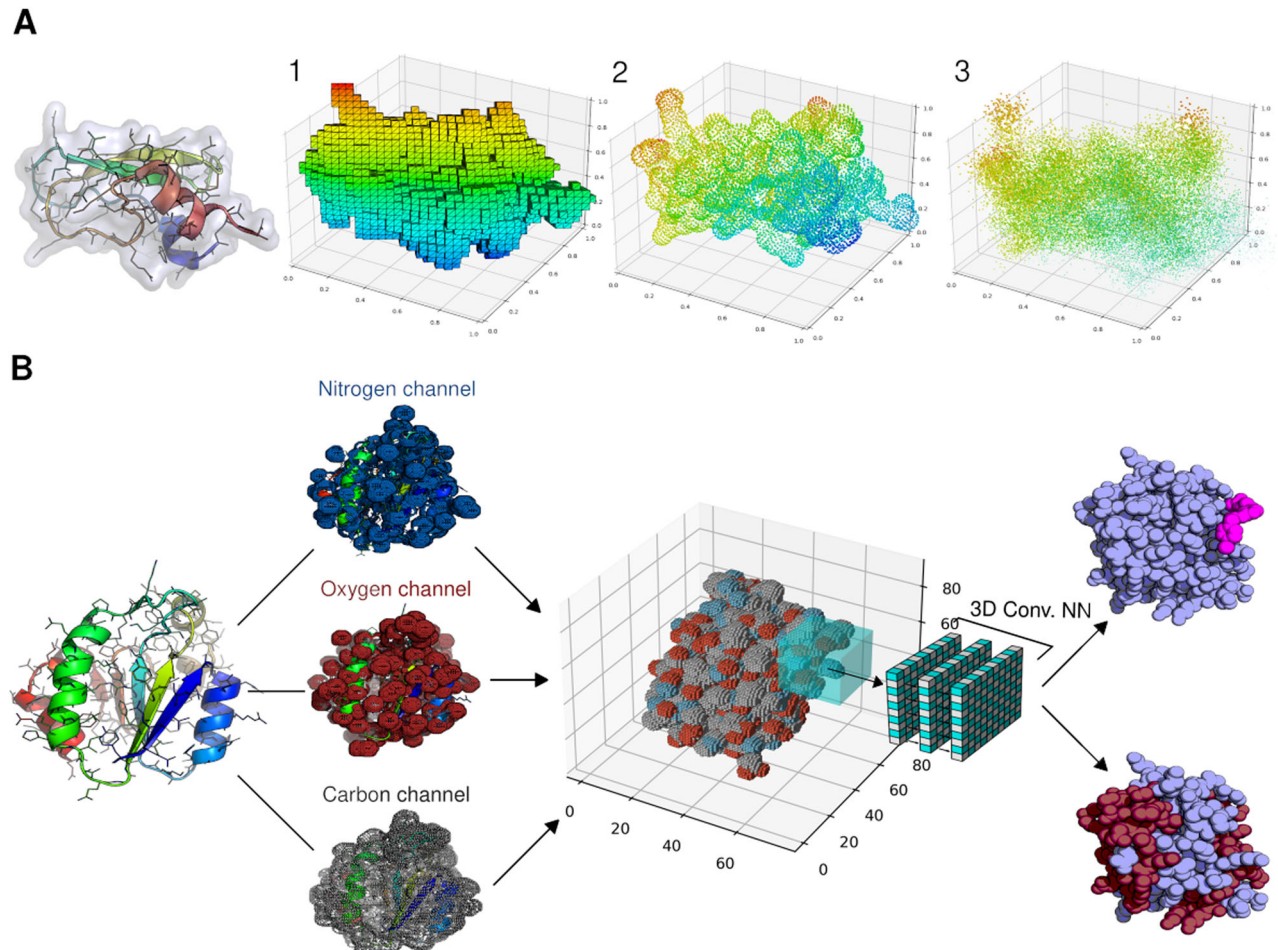

**Fig. 1 Overview on the functionalities of PyUUL. A** Different 3D data representations available using PyUUL of bovine pancreatic trypsin inhibitor (PDB ID: 1BPI (https://doi.org/10.2210/pdb1BPI/pdb)). From left to right: voxel representation (Subfigure 1), surface point-cloud representation (Subfigure 2), and volumetric point-cloud representation (Subfigure 3). **B** Schematic illustration of how the multichannel voxel representation works using PyUUL. The input 3D structure of a protein is represented in a voxel grid, where each channel shows the occupancy of nitrogen, carbon, and oxygen atoms. This 3-channel voxel representation can be used as input for a 3D convolutional neural network that iteratively samples the entire protein volume and processes this data to address problems such as protein pattern recognition or protein-ligand interaction learning.

In this context, PyUUL can transform the protein structures to 3D volumes that can then be handled by NN modules, such as 3D convolutional NN[30].

As an example of how NN models can be used with PyUUL, we built a NN to recognize alpha helices within a protein structure. We chose this example because, while it is sufficiently easy to be solved by a relatively simple NN, it nonetheless clearly requires the extraction of structural information from the voxelized volume describing the input proteins, showing that the protein is correctly transformed into a tensorial data structure and structural information is preserved. Moreover, the NN is sufficiently simple to allow a clear visualization of its learning process. Figure 1B shows an overview of the steps performed to build a volume-based NN using PyUUL. Figure 2A and Supplementary Movie 1 show the evolution of the 3D convolutional NN during training. The voxels highlighted in gray are the ones that the NN, at each specific learning stage, recognizes as part of an alpha helix. As shown in the figure, during the increasing training epochs, the NN rapidly converges to the identification of the alpha-helix regions. The purpose of this example is to show that PyUUL's volumetric representation is highly informative and that ML methods can easily model structural biology concepts (e.g. secondary structures) from it.

The tensorial representation provided by PyUUL is the standard input of several computer vision utility libraries such as torch-Geometric[31] and torch-points3d[32]. The first one provides bleeding-edge network modules such as graph-CNN, while the second one also includes several network architectures that can be used in a plug-and-play fashion and, if paired with PyUUL, in a completely out of the box manner.

We provide a tutorial for this example at: https://pyuul.readthedocs.io/examples/example1.html

**PyUUL works also with nucleic acid and small molecules.** PyUUL has been built to be as general as possible. Therefore, it is possible to convert any biological molecule to tensorial representation, including small molecules and nucleic acids. This feature has the purpose to help the user to apply modern ML algorithms in topics such as drug repositioning[33] or chemical discovery[34]. Moreover, the possibility to handle nucleic acids can be useful when dealing with super-molecular complexes or ribozymes. The current version of PyUUL provides the data of most of the common non-protein biological molecules and atoms. However, it can also deal with molecules containing more exotic atoms if their atomic radius is provided. A tutorial on how

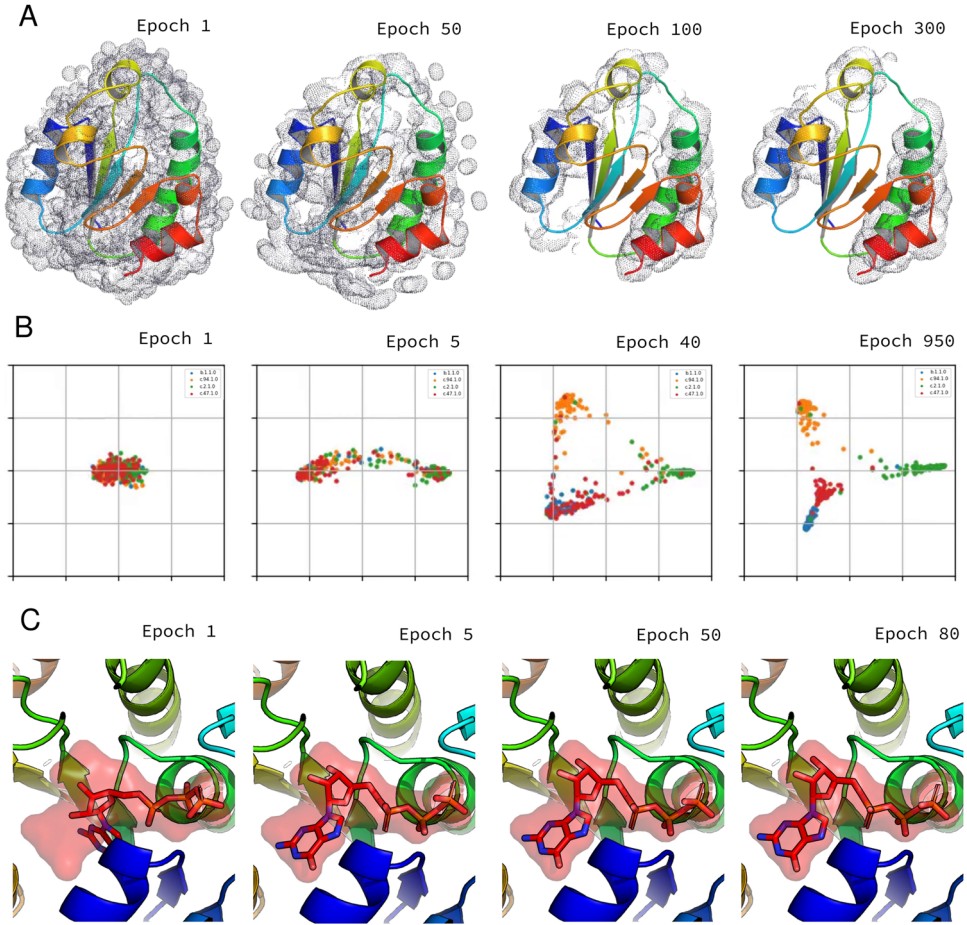

**Fig. 2 Application of PyUUL to structural bioinformatics problems. A** The learning process of a 3D convolutional neural networks on a structural biology task: the picture shows the training procedure of a neural network on a voxel representation of a protein. The network iteratively learns to recognize alpha helices. The voxels that are predicted to be part of an alpha helix at each leaning stage are highlighted in gray. Supplementary Movie 1 shows the complete evolution of the network during training. **B** The learning process of a 3D Siamese NN trained with metric learning. The network assigns a 2D vector (two latent features) to every protein in accordance to its 3D representation obtained with PyUUL. Each point represents a protein and each color a different fold. With the training, it optimizes this signature in a way that proteins belonging to the same protein class have a similar signature. Supplementary Movie 2 shows the complete evolution of the network during training. **C** Optimization of the conformation of GTP. The figure shows a neural network optimizing the conformation of GTP in its binding pocket. This is done first training a network to recognize original GTP-protein complexes from the ones in which GTP has been randomly rotated, defining a scoring function that describes how likely the protein-ligand complex is to be the optimal one. The parameters of the network are then frozen and the network is asked to reposition GTP to the right position in the pocket of their respective protein. This protein-ligand optimization can be done simply calculating the gradient of the scoring function with respect to the GTP coordinates. Supplementary Movie 3 shows the complete evolution of the network during training.

to generate volumetric representations of unsupported molecules is provided in PyUUL's documentation.

**End-to-end structure-based protein clustering with PyUUL.** Novel NN approaches often blur the division between regressors and classifiers, since their flexibility allows these models to address intermediate flavors of prediction tasks. One example of this is metric learning[35], a technique that allows NNs to learn how to map input objects into a learned latent space in which a target definition of distance between these objects (metric) is preserved. This is for example used for one-shot learning in image recognition[36]. Specific NN architectures, such as Siamese network, have been introduced to solve these tasks[37]. Siamese networks are a type of network architecture that takes an input and provide a latent embedding or a "signature" of the input. The output of the network is abstract and has no direct biophysical meaning. However it acquires specific properties when compared with the embeddings obtained from other inputs (i.e. the euclidean distance between two embeddings).

Another common aspect that is shared between computer vision and ML on 3D biological structures is that the models used should be rotation invariant to recognize objects regardless of their rotation (i.e., the angle at which the picture has been taken). To overcome this problem various solutions have been proposed, and among the most recent, an invariant architecture called Spatial Transformer (ST)[38] has been introduced. STs are modules that learn to perform an instance-specific affine transformation of the inputs, thus allowing the NN to learn the optimal set of transformations (e.g. rotation, scaling, shear, etc.) to extract information from each input instance, thus making the model more robust to perturbations. In the following we show that these solutions are directly applicable to 3D structures within PyUUL's framework.

As a proof of concept showing the capabilities of PyUUL, we propose a NN for rotation-invariant metric learning on protein structures for protein fold recognition. We collected 491 protein structures divided in 4 different SCOP[39] classes and we trained a NN to perform a supervised clustering based on the PyUUL

volumetric protein representation. The NN we used is a 3D Siamese NN[37] that takes as input the PyUUL voxel representation of the proteins and maps them into a latent space, using the ST layers[38] to ensure rotation invariance. This latent space is an abstract vectorial representation of the proteins, where proteins with similar structures are represented by similar vectors, while proteins with completely different structures are more distant (see Methods for more details). In other words, we built a supervised clustering algorithm that assigns a signature to proteins based on the shape information present in the PyUUL volumetric representation used as input.

The learned latent space is 5D, thus we performed Principal Component Analysis, a linear method for dimensionality reduction based on rotation of the reference frame to maximize the variance of each axis, in two dimensions to allow the visualization of the trajectories of the points representing the input proteins during training (see Fig. 2B and Supplementary Movie 2). A 5D encoding was chosen since the network had convergence problems with a smaller embedding. We can see that, in this latent space, the points representing proteins belonging to different classes are moved away from each other, while proteins from the same class are moved closer together, ending up with 4 clusters corresponding to the training classes. Since the NN learns distances in a metric space, the positions on the plot are arbitrary, given that the distances are preserved. Metric learning with PyUUL can be used to cluster proteins accordingly with any structure-related global characteristic of proteins. More details about the results of the proposed network can be found in Supplementary Note 3.

**Gradient-based protein structure optimization with PyUUL.** An important problem in structural biology is the optimization of a protein or a drug using some predefined objective function, such as a force field. This is the case of the force field-based optimization that can be performed by Rosetta[40] or FoldX[41], in molecular dynamics simulations, in docking and in loop modeling.

Let us suppose that in our optimization task the objective function is provided by a differentiable force-field $F$ and that the target biological structure (i.e. a protein) is represented by its 3D coordinates $\mathbf{X}$. $s = F(\mathbf{X})$ thus provides a score $s$ of the quality of the global structure. In the case of a force-field, this score would be a $\Delta G$ and the highest score conformation would be the one with the minimum energy.

However, the objective function $F$ might be any differentiable function, including for example one derived from an arbitrarily complex NN that has been trained to recognize the minimum energy states of a specific class of biological structures. In this case, the score would not have a direct physical meaning, but its minimization would have the same effect as $\Delta G$ protein optimization.

The volumetric representation provided by PyUUL is fully differentiable and this means it can be used as an internal step in a gradient-descent optimization. More specifically, we can calculate the gradient $\partial F(\mathbf{X})/\partial \mathbf{X}$ of the score $s$ with respect to the input coordinates $\mathbf{X}$. We can then use the standard gradient-based optimizers provided by Pytorch (that are usually used to train NNs) to optimize the input structure $\mathbf{X}$ by moving each atom in the direction that maximizes (or minimizes) $s$. Thanks to flexibility of Pytorch autograd mechanism[18], this is possible in just a couple of lines of code. Alternatively, as suggested by AlQuraishi[42], it is possible to compute $\partial F(\mathbf{X})/(\partial\phi\partial\psi)$, thus differentiating $s$ with respect to the torsion angles $\phi$, $\psi$, so that the optimization of the position of the atoms natively preserves the bond distances.

To show that this kind of end-to-end optimization is possible with PyUUL, we tackled the task of optimizing guanosine triphosphate (GTP) structure in its binding pocket. GTP is a molecule that plays a crucial role in many regulatory processes, often promoting a conformational change,[43]. The orientation of the ligands, however, is often troublesome for standard docking algorithms[44]. We thus used PyUUL to build a NN to learn the optimal conformation of the protein-GTP complex and then used it to dock the ligand into the pocket. To do so, we first collected 515 proteins in complex with GTP (available in the git repository). For every protein, we rotate the GTP at a random angle, and we train a 3D convolutional NN to recognize the rotated conformations from the original ones. After the training, we give to the NN a protein-ligand complex in which the GTP has been randomly rotated and translated, and we use the NN to put it back to the optimal place, effectively performing an optimization of the GTP-protein binding. The whole procedure has been performed in grouped 5-fold cross-validation, in which the intergroup sequence identity is at most 30% in order to prevent the network from learning just the amino acidic composition. The proteins involved in the classification are therefore non homologous to the one on which the GTP pose optimization test is performed. 473 out of 515 of the optimized complexes converged to a solution, while 42 did not converge after 100 iterations. Figure 2C and Supplementary Movie 3 show the iterative refinement of the GTP position performed by the volume-based NN for 5H74 (https://doi.org/10.2210/pdb5H74/pdb). More details about performances and validation can be found in Supplementary Note 3 and Supplementary Fig. 2.

**Protein signature encoding with point clouds.** In previous sections we described possible applications of PyUUL in docking and supervised clustering using pure volumetric ML techniques. In this section we want to merge these two concepts and generate fixed-length signatures for each GTP binding pocket. Such embedding could be used, for example, for in-silico drug screening, similarly to what has been done with molecular fingerprints for small molecules. We therefore want to find an encoding that preserves the similarity between pockets. To do so, we are going to use surface cloud point representation. The network architecture, however, could be used with volumetric point clouds as well. First, for each GTP binding protein under scrutiny we calculated the surface point-cloud representation using PyUUL. We then used the Fast Global Registration algorithm, an algorithm often used for 3D point clouds registration in computer vision which finds the optimal superimposition between two point clouds, to estimate the pairwise distance of each pair of pockets. We then applied a variant of the encoder FoldNet proposed in ref. [45]. The network takes as input a cloud point and it generates a signature of a fixed size. For more details about the network architecture refer to the methods section. Differently from protein recognition section in which a label (protein class) was assigned to every protein, here we only have pairwise distances. Metrics learning and contrastive loss are therefore not directly applicable to the problem. In order to obtain the pocket signatures, we propose a regression-based approach: we calculate the signature of each pocket, we then calculate a pairwise distance matrix between them and then we compare it with the one obtained with the fast global registration algorithm. The network therefore learns to assign signatures that preserve the distance matrix of the point clouds. We tested this approach on the GTP binding pockets used in previous section performing a 5-fold stratified cross-validation in which proteins with SI greater than 30% are grouped together. Using 10 features as encoding dimension we could reach a Pearson's correlation

coefficient between the signature based and the cloud point based pairwise distance matrix of 0.72. More information about the network can be found in Supplementary Note 3 and Supplementary Table 1.

## GPU parallelization, sparse tensors, and resources usage.
PyUUL can calculate the volume using sparse tensors from Pytorch. This often provides a boost in performances, in particular when many volumes are batched together. This results from the fact that empty voxels are often predominant. Even if currently Pytorch support for sparse NNs is somehow limited, many third-party libraries alongside with the Facebook research group are currently been developed to allow sparse computations[46]. PyUUL has full GPU calculation support, and all the operations can be handled by both the CPU or the GPU. Supplementary Fig. 1 shows a benchmark of the speed performances and RAM requirements.

## Discussion
In structural biology, the adoption of the latest deep learning algorithms has been slower than in other fields of science. The main factor has been that biological structures are stored in peculiar data formats, which are not trivial to use as inputs for deep learning methods. We developed PyUUL to solve this problem by connecting the raw structural data contained in macromolecules with the input formats required by deep learning algorithms (i.e., NNs).

There are other publicly available packages that can convert biological structures to volumetric (mainly voxel-based) representations[20,47]. However, in PyUUL the volumetric transformation is performed in a differentiable way, meaning that the gradient calculated by Pytorch can pass through it. This feature is essential for the development of end-to-end sequence-to-structure NN, since the gradient is supposed to flow from the input to the loss function without interruptions.

The recent success of AlphaFold[48], that has an end-to-end architecture, in the CASP14 competition highlighted the potential of this type of NN. PyUUL can therefore be used to easily implement 3D-based NN modules in such architectures.

PyUUL is an open-source software, which will continue to be updated and expanded based on community feedback and the advancements in the field.

## Methods
**PyUUL implementation**. PyUUL is implemented in Pytorch[18]. The autograd function of Pytorch allows the automatic propagation of the gradients between subsequent mathematical operations and it natively supports GPU computation. In the following sections, we describe the steps required to transform biological macromolecules structures in volumetric entities.

**Processing biological structures with PyUUL**. PyUUL has been designed to be as transparent as possible to the user. It provides some basic utilities and parsers to deal with standard PDB files. `utils.parsePDB` takes as input a single PDB file or a folder containing PDB files and returns a tensor with the coordinates and a list of atom names. The atom names can therefore be used to create channels (`utils.atomlistToChannels`) and a tensor of radius (`atomlistToRadius`). These two functions provide the standard channels and atoms radius of PyUUL, but the user can redefine both these tensors in order to change (1) how atoms types are grouped together and (2) the radius of each atom type.

**Voxel-based representation**. In the following section, we describe how the voxel representation of the biological structures is obtained.

To translate a PDB structure into a voxelized volume, the first step is to define the volume's boundaries. PyUUL starts by building a parallelepipedic box around the input structure and divides it into voxels. Every voxel has a side length that can be defined by the user (default = 1 Å), which determines the resolution of the output 3D representation of the volume. When multiple input structures are given as input (coordinates and radius with batch greater than 1), the box is defined in a way that it contains all the different structures. This can be seen as 3D padding, and it allows a fast and effective batching of the tensors.

PyUUL estimates the volume occupied by an atom as a function of its radius. Atoms are therefore not simply point-like entities, but sphere-like solid objects. This occupancy function needs to be differentiable to allow Pytorch to propagate the gradient in the subsequent calculations. Operations that requires gradient calculation are performed using pytorch 1.9, while the others are done using numpy[49] 1.19. Scikit-learn 1.0.1[50] is used for scaling procedures. PyUUL provides two volume density functions, that slightly change the definition of the volume associated to each atom. The user can decide which function to use passing the corresponding parameter to the Python object (see online documentation for more information). The first one is a sigmoid-like function, as shown in Eq. (1)

$$V_a(d, r_a) = \frac{1}{1 + e^{s(d - r_a)}} \qquad (1)$$

where $V_a(d, r_a)$ is the fraction of volume occupied by atom $a$ of radius $r_a$ at distance $d$ by its center, and $s$ is a steepness parameter (default = 10) that defines how fast the atom occupancy decreases at distances greater than their radius. In more practical terms, this function defines the occupancy of an atom with a sigmoid that has the inflection point in correspondence of the point in which the distance is equal to the radius of the atom.

The second one is the one defined by Li and coauthors in ref. [51] and described in Eq. (2)

$$V_a(d, r_a) = exp\left(-\frac{d^2}{\sigma^2 \cdot r_a^2}\right) \qquad (2)$$

To compute the final volume occupied by the atoms, we need to change the reference system from the atom point of view to the voxel one. We thus need to integrate over every voxel in the box. In other words, for each voxel we have to sum the occupancy generated by every atom. To reduce the computational requirements of the library and given the fact the contribution to the volume of an atom decreases exponentially for distances greater than the atom radius, every atom at a distance greater than 10 Å from the point in the volume is considered to have contribution equal to 0. This value can be adjusted by the user. For the sigmoid-based occupancy function, the occupancy of a voxel is defined as the sum of the contribution of each atom to the point in the center of the voxel (Eq. (3)).

$$VOXEL_{i,j,k} = min\left(\sum_{n=1}^{N_a} V(||\mathbf{C_{i,j,k}} - \mathbf{x_n}||, r_{a_n}), 1\right) \qquad (3)$$

where $N_a$ is the number of atoms in the channel under consideration, $\mathbf{x_n}$ are the coordinates of Atom $n$, $r_{a_n}$ is the radius of Atom $n$, $i$, $j$ and $k$ are three integers that identify the coordinates of the voxel in the box and $\mathbf{C_{i,j,k}}$ is the center of the voxel $i$, $j$, $k$.

For what concerns the Li's occupancy function, we used Equation 2 of ref. [51] as shown in Eq. (4):

$$VOXEL_{i,j,k} = 1 - \prod_{n=1}^{N_a} V(||\mathbf{C_{i,j,k}} - \mathbf{x_n}||, r_{a_n}) \qquad (4)$$

For both functions, every channel is considered as independent and the atoms belonging to different channels do not interact nor share volumetric contributions.

**Surface point cloud**. The surface point cloud is generated in two steps: in the first one, points are sampled around each atom using a Fibonacci lattice on a sphere of the radius of the atom. The usage of Fibonacci lattice ensures that the points are sampled homogeneously on the surface of the sphere. We than calculate the volumetric occupancy of each sampled point using Eq. (4). By contrast with the voxel representation, we investigate the occupancy on the sampled points instead of the center of the voxel. We therefore remove all the points that have a volumetric occupancy greater than 0.5, removing the part of the atoms' surface that is buried by other atoms. To have a constant number of points per structure in a batch, we randomly sample a number of points (defined by the user, default 5000) from each structure.

**Volumetric point cloud**. Differently from the surface point cloud, in the volumetric point-cloud representation, we do not have to deal with occupancy, since this representation has the goal to sample points inside the protein's volume. For each atom, we therefore randomly sample points with a multivariate 3D Gaussian probability distribution with mean equal to the atom coordinate and $\sigma$ equal to the radius of the atom. As in previous case, we randomly sample a number of points (defined by the user, default 5000) from each structure to obtain batches with constant number of points.

**Neural network for alpha-helix identification**. The goal of this network is to show that a NN is capable of converging to a meaningful solution using a voxelized representation of a protein. For this purpose, we used a single protein (PDB id: 1WOU (https://doi.org/10.2210/pdb1WOU/pdb)) since this protein has both alpha helices and beta sheets. We parsed the protein structure using the `utils.parsePDB` function, obtaining its coordinates and the atom names. We then used the functions `utils.atomNamesToRadius` and

`utils.atomNamesToChannels` to obtain the atomic radius and the channels respectively (with default parameters, see Supplementary Note 1 for more information). We then generated the voxel representation of the protein calling the `VolumeMaker` object.

The NN consists of 3 layers of 3D convolutional modules (with respectively (19, 10, 10) channels) followed by 3-layer feed-forward modules (with (100, 100, 1) neurons). See Supplementary Material S2.1 for more details about the NN. The labels on which the network has been trained were a 3D tensor, assessing if every pixel contained an alpha helical region (positive label) or not (negative label). The network was trained for 200 epochs using binary cross-entropy as loss function with the Adam optimizer and learning rate of 0.01. The code is available as a tutorial in the library documentation.

**Network for gradient-based protein structure optimization**. For what concerns the GTP optimization, we started selecting 515 proteins in complex with GTP from PDB[6]. The complete data set is available in the PyUUL git repository. We calculated the voxel representation of each complex using the functions described in previous section with a resolution of 1 Å. However, in this case we used the reduced number channel hashing (see Supplementary Note 2 for more information about channel hashing). We therefore used 4 channels for the protein atoms and another 4 for the GTP, with a total of 8 channels. To let the network focus on the binding pocket, we selected a cube of side 11 voxels, with the $O'$ atom of GTP in the central one, as already done by other authors[44].

We then built an ST with self attention. The self-attention approach for data structure of variable dimensions has been described in refs. [4,52]. The ST has the goal of ensuring invariance to translation, scale, and rotation. It is required since PDB structures have an arbitrary orientation and the NN predictions should not be influenced by it. A deeper description of STs can be found in ref. [38].

The NN is thus composed of an ST layer, followed by the self-attention module[4,52], which allows the NN to deal with inputs of variable lengths.

The last part of the network is a 3-layer feed-forward module with ReLU activation, layer normalization, and dropout. The final output is given as a single neuron with sigmoid activation. For more information about the architecture, see Supplementary Note 1.

For every training and test set in the 5-fold cross-validation, we perform the following steps:

- Rotation: for every GTP pocket in the train set assign a positive label (1) to the original complex conformation and a negative label (0) to a random rotation centered in the $O5'$ atom of the GTP.
- Training: We train the network described above for 100 epochs to discriminate rotated complexes from original complexes using binary cross-entropy as loss function.
- Optimizing: For every complex in the test set, we randomly rotate and translate (translation sampled from a 3D multivariate normal distribution) the GTP. We then evaluate the transformed complex with the trained NN, we calculate the gradient with respect to a transformation matrix (rotation and translation) and we perform a gradient-based optimization finding the complex that maximizes the output of the network. In order to limit the numerical issues of gradient vanishing, the optimization is performed starting from 10 random transformations of the input. In other words, we reposition the GTP finding the conformation in which the trained network gives the highest "correctness" score.

**Siamese network for end-to-end structure-based protein clustering**. To assess the capability of using PyUUL to compress macromolecules structural information in a meaningful way, we selected SCOP families with more than 100 members that did not belong to the same SCOP fold. We therefore selected 419 proteins belonging to 4 families. We decided to focus on a small number of families to easily visualize the output in two dimensions. We split the families in train and test subsets (80% train and 20% test) and we used the voxelized representation offered by PyUUL (resolution of 2 Å) to feed a Siamese NN and perform supervised clustering.

The network architecture used to perform the supervised clustering of proteins is the same as for GTP optimization, with the exception that it has two neurons as output. The network is trained using a contrastive loss from the Pytorch metric learning library[53]. Contrastive loss train the network to give similar encodings to proteins belonging to the same cluster and different ones to the ones belonging to different clusters.

Then we trained the network for 2000 epochs, using the Adam optimizer with a learning rate of 0.01. Protein encoding has been performed in 5 dimensions to get fast convergence, but it is an arbitrary parameter of the network, which can be set by modifying the number of output neurons.

**FoldNet for GTP binding pocket signature encoding**. In order to show an example of usage of the cloud point representation provided by PyUUL, we built a network that compresses structural information of GTP binding pockets and encodes it in a fixed-size array. As dataset, we used the dataset used in "GTP-protein conformation optimization" section. However, form the 515 proteins, we removed 78 proteins that had multiple GTP binding pockets, obtaining a dataset of 437 pockets. Using a pymol[54] script, we removed the atoms not belonging to the binding pocket (distance threshold of 5Å). Using PyUUL's object `Point-CloudSurface`, we generated the point cloud of every pocket, removing all atoms not involved in the binding of the GTP (based on a distance threshold of 5Åform any GTP atom) and we obtained their surface point-cloud representation using PyUUL. Using the fast global regularization algorithm implemented in torch-point3d[32] library and proposed by Zhou and coauthors[55], we superimpose every pocket pair and we estimate the pairwise distance using the MSE of the matching points using the `get_match` function of torch-points3d. The pairwise distances are the labels of our regression problems that the network will learn from. The network architecture is a variant of the FoldNet encoder developed in ref. [45]. This type of architecture is well known in computer vision and, iteratively applying 1D-convolutional filters and pooling to the input cloud point, provides a permutational invariant (the order of the points of the cloud does not influence the learning procedure) prediction. The NN maps the cloud point in an arbitrary number of dimensions that represents an abstract description of the object under scrutiny. The only difference between our architecture and Tao's network is the presence of an additional 3-layers feed-forward sub-network with 10 each neuron on top of the encoder and the fact that all activations have been changed from ReLU to hyperbolic tangent. All the other parameters have not been changed. A detailed description of the original foldNet network can be found in the relative article[45]. Training and testing has been performed in a 5-folds stratified cross-validation setting, grouping together proteins with a sequence identity higher than 30% . The network has been trained for 30 epochs with Adam optimizer, learning rate of 0.001 and batch size of 50.

**Computational resources requirements**. The speed and RAM consumption performances calculation have been performed on a data set of 100 proteins of size between 100 and 200 amino acids. The data set is available in the Git repository. The speed tests have been performed on an Intel i7-8850H CPU and an Nvidia P1000 GPU.

**Reporting summary**. Further information on research design is available in the Nature Research Reporting Summary linked to this article.

## Data availability
The PDB structures used in this study are available at https://bitbucket.org/grogdrinker/pyuul. All the structures have been downloaded from the Protein Data Bank (PDB)[6]. Source data are provided with this paper.

## Code availability
A Python implementation of the algorithm is available at https://bitbucket.org/grogdrinker/pyuul/. It can also be installed via the `pyuul` PyPI (https://pypi.org/project/pyuul/) or conda package (https://anaconda.org/grogdrinker/pyuul).

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

## Acknowledgements
The Switch laboratory is supported by the Flanders Institute for Biotechnology (VIB, grant no. C0401); KU Leuven (postdoctoral mandate PDM/19/157 to G.O.); and The Research Foundation – Flanders (FWO, project grants G053420N and SBO S000722N). D.R. is founded from an FWO post-doctoral fellowship. The authors are grateful to Teresa Izuel Idoype (Stem Cell and Developmental Biology, KU Leuven) for designing the logo for the PyUUL tool.

## Author contributions
G.O. developed the algorithm. G.O. performed tests and validations. G.O., D.R., R.D.-R., Y.M., J.S., and F.R. wrote the article.

## Competing interests
The authors declare no competing interests.
