## [Peer Review File · Nature Communications]

PyUUL provides an interface between biological structures and deep learning algorithmsReviewers' Comments:

Reviewer #1:

Remarks to the Author:

The authors report a novel python package called "PyUUL" for generating deep learning features for 3D protein structures. The package provides three different representations of the macromolecules including voxel, surface and volume point cloud to enable direct deep learning of protein structure using common machine learning frameworks. They further demonstrate the utility of PyUUL through three case studies including fold recognition, docking and structural clustering. Although these representations have been previously reported, I think the package could have potential impacts in broadening the use of deep learning for structural bioinformatics as there are few dedicated packages like this publicly available. Overall, the package shows some promise and is unique but at the current stage the result does not appear to be sufficiently convincing and further improvements in paper presentation of the analysis and results is necessary to appeal to a broader audience.

The overall presentation of the manuscript:

1. The figures should be better organized. Figure 1A and B discussed several representations of the proteins but then Figure 1C showed GTP docking. I think Figure 1C could move to another main figure to expand on the results and analysis. Figure 1 is best reserved for the general schematic of the approach.
2. Figure 2 shows three embedded videos within the figure. Should these be figures instead? There are a couple of places where the results were conveyed via video like PCA (video 23), which could have a figure representation for the main text as well.
3. There are no analysis/result figures for the fold recognition section using NN.
4. There are no analysis/result figures for the structural clustering section using NN.
5. There are a few figures for GTP docking results in 1C and should be expanded.
6. Under the heading "End-to-end structure-based protein fold recognition with PyUUL", unsupervised clustering was discussed instead. A short description of fold recognition of alpha helix was found in the method section but not in the result section.
7. To better organize, I would advise having a dedicated section for the feature presentations as well as each of the three case studies with accompanied figures/texts showing the analysis/prediction results in the main and detailed methodology description in the method section.
8. I would also advise having an in-depth discussion section. The authors should expound further on the strength and limitation of their approach as well as future direction.

Methodology:

1. How are different representations compared in terms of utility and performance? In the case studies, only voxel representation is used without giving explanations of why this particular representation is chosen. I think more detailed characterizations of each representation would help the readers choose the best representation for their applications.
2. Figure 1C, I would recommend reporting the AUC values of the ROC curves.
3. The alpha helix recognition section under the Neural network example is vague. It is unclear how the model was trained, what training datasets were used, how the model was validated and what predictions were made. I ran the python package and it appears that the section in the main text was a tutorial from the package and therefore I think the scope is somewhat limited for the main text. This is not a complete study and I would suggest expanding on this to showcase the full fold recognition prediction using PyUUL.
4. Siamese network for supervised clustering with metric learning section has more description of the models but lacks results.
5. NN-based GTP docking is the section with better coverage. The models built for each of the three case studies should be more rigorously defined, validated and the analysis/prediction result should all be reported.

6.I was able to run the python package completely without significant issues.

NN-based GTP Docking:

- 1.To improve superiority, perhaps can compare the pose scoring performance with 1 or 2 existing pose scoring functions in current docking programs like DOCK, AutoDOCK etc.
- 2.The pose prediction result seems good but is done for 1 GTP-bound complex only. To prove generalizability, I would recommend evaluating additional complexes to ensure that the approach will work on any protein-ligand complexes.
- 3.How do the performances differ with respect to protein subtypes and structures? Similarly, how are the 515 protein structures split between training and validation?
- 4.Low (<30%) sequence identity does not guarantee low binding sites similarity. I would more rigorously annotate the binding site in the dataset in terms of their local 3D structural similarity. One interesting application that the authors can potentially explore is to use the PyUUL to compare binding pockets and then correlated that to the RMSD values to the original pocket. Using other published binding site comparison algorithms is also a possibility.
- 5.The PyUUL framework applied on GTP-protein complex seems like an interesting application as it opens door to potential applications of NN for drug design and optimization. My main concern is the potential applicability of the framework for novel ligands without a known co-crystal complex. I think that very limited training data size for most of the ligands could potentially diminish the performance and the applicability of the framework.
- 6.Similarly, could the approach be applied to GTP analogs docked to the same site? or different GTP conformations considering ligand flexibility?
- 7.“For every protein, we rotate the GTP at a random angle, and we train a 3D convolutional NN to recognize the “wrong” conformations from the original ones.” Were there only one or multiple sampled conformation? It is still possible that a random rotation could still generate a “close-to-correct” conformation if the rotation is small unless an RMSD cutoff was used.
- 8.“we ask the NN to put it back to the optimal place, effectively performing an optimization of the GTP-protein binding.” I don't think this is an accurate description. The NN does not predict transformation but only predicts the score for a particular transformation.

Minor comments:

There are multiple typos found throughout the text:

- 1.Figures 1C 3 and 4 shows... -> show
- 2.More information are available in Supplementary Figure S1 -> is available
- 3.The NN consists in 3 layers of 3D convolutional modules -> consists of
- 4.The NN is thus composed by a spatial transformer layer -> composed of
- 5.313 out of 515 of the optimized complexes had a RMSD lower than 2 with respect to the original crystal structure -> an RMSD lower..
- 6.but its minimization would have the same effect of ΔG protein optimization. -> as ΔG protein optimization.
- 7.An important problem in structural biology is the optimization of a protein or a drug in function of some predefined objective function, such as a force field. -> using some predefined...
- 8.This protein-ligand optimization can be done simply calculating the gradient of the scoring function respect to the GTP coordinates. -> with respect to the GTP coordinates.
- 9.The input 3D structure of a protein is represented in a voxel grid, where each channel show the occupancy of nitrogen, carbon, and oxygen atoms. -> shows

Reviewer #2:

Remarks to the Author:

The authors present a python library named PyUUL to allow biological structures in PDB format to be converted to a format suitable for deep learning libraries, within PyTorch. They describe the usage of this library on a few sample problems.

My understanding from the paper is that the main contribution of the paper is software to convert the PDB representation into one of three representations (voxel-based, point-cloud surface, volumetric point-cloud). If my understanding is correct, I would say that the contribution is modest, but useful. Perhaps more detail about the input PDB format and the transformation algorithms could be included.

It is not clear how comprehensive this capability is. Perhaps the authors could provide in a table all of the deep learning techniques that can be utilized as a result of their work. I am also unclear whether this package has been used by other researchers besides the authors. If the authors can show this, that would be a big plus.

Finally, the paper is readable, but the clarity could be improved, perhaps with the help of a professional proofreader.

Reviewer #3:

Remarks to the Author:

Orlando and co-authors present a library that allows to process 3D information of protein and small molecules and to make it "machine-readable" and suitable for modern end-to-end deep learning algorithms. This is very important given the recent progress made by AI in the field of protein structure prediction, synthesis prediction and drug design. The open-access character of the platform and the promised flexibility make it promising for accelerating the impact of AI in the life sciences, in a multitude of structure-based tasks. The paper is well written and very clear in several technical explanations, which make it accessible to a broad readership. I only have a few comments to further improve its readability and accessibility to less expert readers.

Major comments

- Gradient-based protein structure optimization with PyUUL: it is not clear what authors mean with "wrong" conformations. Please consider improving the description of the goals and the procedure.
- Overall structure: Some elements of the presented platform are just mentioned in the general introduction and then left in the methods (e.g., surface point cloud). This might be misleading for readers and "hide" important features. I suggest putting the more "generic" description of such representations in the main text, while still leaving the technical details in the methods section.
- The authors mention the applicability of their pipeline to small molecules as well, but this part has not been really explained in the manuscript. There is also a lack of references to the recent advances in the field of AI for small molecules. The authors could consider expanding upon such statements to improve the paper clarity and provide a reasonable overview of the real possibilities of the proposed library.

Minor comments

- Briefly explain the concepts behind Siamese networks at the first mention for less expert readers
- Briefly explain the concepts behind spatial transformers at the first mention
- Why is the learned latent space 5D? Consider explaining it at the first occurrence
- Explain what a PCA is
- Define all acronyms at the first occurrence (e.g., GTP, NN, PCA, pyUUL [if any])
- In addition to AlphaFold, also consider citing this paper as well: Baek et al. Accurate prediction of protein structures and interactions using a three-track neural network. Science. 2021 Jul 15.

REVIEWER COMMENTS

Reviewer #1 (Expertise: computational chemistry):

Before addressing the comments of reviewer #1, we would like to stress a crucial point about the paper. All the Neural Network (NN) models presented have purely illustrative purposes. Their goal is indeed to show how PyUUL library could be used and we do not claim by any means that these minimal models are better or comparable with state of the art methods addressing similar tasks. The limited scope of the examples we show have the goal to allow the reader to focus on the usefulness of the library and to show the reader which classes of structural bioinformatics problems can be addressed with it.

The goal of PyUUL is therefore to relieve data scientists from any difficulties while managing 3D models of biological molecules (e.g. protein structures), allowing them to focus solely on the data analysis aspects.

The authors report a novel python package called "PyUUL" for generating deep learning features for 3D protein structures. The package provides three different representations of the macromolecules including voxel, surface and volume point cloud to enable direct deep learning of protein structure using common machine learning frameworks. They further demonstrate the utility of PyUUL through three case studies including fold recognition, docking and structural clustering. Although these representations have been previously reported, I think the package could have potential impacts in broadening the use of deep learning for structural bioinformatics as there are few dedicated packages like this publicly available. Overall, the package shows some promise and is unique but at the current stage the result does not appear to be sufficiently convincing and further improvements in paper presentation of the analysis and results is necessary to appeal to a broader audience.

Thank you for your comments. We are aware that the examples we show in the paper have already been subject of research. Those examples have the purpose of showing how our library could easily allow data analysis and Machine Learning (ML) on structural biology problems, with particular focus on algorithms from computer vision, while masking the inner complexity of protein structure representation.

The application of computer vision algorithms to structural bioinformatics have already proven their effectiveness. For example, Jimenez et al. (2017) and Pu et al. (2019) addressed protein binding site prediction based on voxelized protein representation. Li et al. (2020) used the voxelized representations for protein stability prediction. As you mention, the goal of the package is to broaden the use of advanced ML in structural bioinformatics by allowing any data scientist to apply cutting edge computer vision algorithms to structural biology data.

Moreover, the fact PyUUL provides **differentiable representations** makes it applicable to end-to-end sequence-to-structure neural networks, a class of neural networks to which AlphaFold [Jumper et al. (2021)] belongs to, making this feature even more valuable after the recent breakthrough in protein structure prediction.

The overall presentation of the manuscript:

1. The figures should be better organized. Figure 1A and B discussed several representations of the proteins but then Figure 1C showed GTP docking. I think Figure 1C could move to another main figure to expand on the results and analysis. Figure 1 is best reserved for the general schematic of the approach.

We split the figure following your suggestion. Now a schematic description of the library is in the multipanel figure of the main paper, while the GTP and computational performances belong to two other supplementary figures. We hope this makes the reading clearer.

2. Figure 2 shows three embedded videos within the figure. Should these be figures instead? There are a couple of places where the results were conveyed via video like PCA (video 23), which could have a figure representation for the main text as well.

We do not really know how the final article layout will handle videos. Nature communication is an “online only” journal, so we were assuming videos could be embedded in the final PDF. However, we are not aware of the policy of the journal about this topic, and we will discuss it with the Editors in due time. Of course, if possible, we would like to keep them as videos since we think they are an effective way to show the evolution of a network. We are, however, open to every option (even transforming them into pictures) and we will follow the indications of the editorial office of Nature Communication.

3. There are no analysis/result figures for the fold recognition section using NN.

The paragraph you are referring to is meant to be a first easy example on how PyUUL works and how the output of PyUUL can be directly used inside a Neural Network (NN). This example is just meant to show that PyUUL data structures can be used directly in a NN and that they contain actual structural information. It is therefore a simple introduction to PyUUL’s input/output data structures that can be followed step by step in the tutorial of the online documentation. We added a sentence to point this out at the end of the paragraph.

4. There are no analysis/result figures for the structural clustering section using NN.

This is the first of the examples of usage that are meant to suggest possible applications to the users of the library. We did not discuss the performances on purpose because we did not want the reader to think the one we presented is an actual method.

However, 3D-based NNs on protein structures for fold recognition have already proven their effectiveness, for example in VoroCNN [Igashov et al. 2021] and, in a similar task, in GraphQA (in which graph convolutional NN are used to assess the quality of a protein model). Moreover, Liu et al. (2021) claims that deep learning will be crucial in future fold/function recognition tools. However, without a library that handles the interface between structure and tensorial data structures, such as PyUUL, this transition to deep learning will be indeed more complicated. To clarify this aspect, we have now added “*proof of concept*” (PoC) in the title of the section.

The idea behind showing these PoCs is also that the readers might find connections between our examples and some of the biological problems they are dealing with in their research, so might use PyUUL in order to develop a similar approach.

We would also like to point out that, as with most clustering algorithms, this method is difficult to objectively evaluate since it is difficult to objectively quantify when a particular clustering is “better”. We, however, understand that the reader might be interested in the results/issues we experienced with our networks, or verify that the results are indeed non-random. We therefore added a section in supplementary material (section S4.1) in which we discuss the results and the problems we had in developing the network.

5. There are a few figures for GTP docking results in 1C and should be expanded.

We moved figure 1C to supplementary, since it might mislead the reader to think that we are presenting an actual docking tool. As mentioned above, this is just a proof-of-concept example to show how PyUUL might be used to expand the classes of architectures that can be applied to biological structures.

The discussion about the biological results (performances of the NNs) is voluntarily avoided because we did not want the reader to be misled and to focus too much on the (simple) NN we developed, because this paper is not about the NN themselves, but it is about a library that bridges the gap between advanced ML methods (e.g. Deep Learning) and structural biology. We therefore think that a too articulate discussion of the performances might make the paper more complicated and might induce the reader to misunderstand the main point of our work. However, as for the clustering PoC, we added some discussion in the supplementary material (section S4.2), focusing on the convergence of the network.

6. Under the heading “End-to-end structure-based protein fold recognition with PyUUL”, unsupervised clustering was discussed instead.

Thanks for the suggestion, we changed the title of the section, making it more consistent. We also made the titles in results, methods and supplementary material more consistent.

A short description of fold recognition of alpha helix was found in the method section but not in the result section.

The goal of the alpha helix recognition example is to show the reader an example of how input/output data structures of PyUUL work. Moreover with the video we want to show that the gradient correctly flows through the volume representation and that a neural network is therefore able to optimize parameters based on it. We understand this might have not been clear in the text and we improved it. Moreover, from a bioinformatics standpoint, we would not consider this example “fold recognition”, but just an exercise showing that a NN put on top of PyUUL can indeed learn structural entities (such as secondary structures) from the data format

provided by it, indicating that structural information is preserved by our library, and just transformed into a ML-suitable format.

7. To better organize, I would advise having a dedicated section for the feature presentations as well as each of the three case studies with accompanied figures/texts showing the analysis/prediction results in the main and detailed methodology description in the method section.

We added a supplementary section (Section S4) in which we discuss the results of the networks. However, as mentioned above, the models we show are proof of concepts, and not actual full-fledged methods. The real results of the paper are the representations and the fact that networks converge, not the performances of the very simple networks we present.

8. I would also advise having an in-depth discussion section. The authors should expound further on the strength and limitation of their approach as well as future direction.

We added a discussion about possible applications in end-to-end sequence to structure NN in the discussion, taking some newly published papers (such as AlphaFold) into consideration and the new perspectives PyUUL provides. Moreover, we discussed the problems (mainly about convergence) we experienced in the development of the example networks in Supplementary Section S4 and how we addressed them.

Methodology:

1. How are different representations compared in terms of utility and performance? In the case studies, only voxel representation is used without giving explanations of why this particular representation is chosen. I think more detailed characterizations of each representation would help the readers choose the best representation for their applications.

This is indeed a difficult question to answer. There is a combinatorial number of choices and variables that must be chosen when developing a NN-based tool. The type of problem under scrutiny, architecture of the network, the number of parameters, the way in which the input is encoded, all influence the model performances. Unfortunately, there is no way to say in advance which is the best representation (this is commonly referred to as “no free lunch theorem” in ML jargon). This is also the main reason why we decided to offer various options to the users when it comes to the ML-understandable formats provided by PyUUL.

The data format must be therefore selected in accordance with the computational resources, the task and the NN architecture.

For instance, if we use a 3D-convolutional-based NN, we have to use a voxel-based representation, since the input needs to be a 3D grid. A point-cloud based representation, on the other hand, is a collection of 3D points, and so permutation-invariant NNs must be used (e.g. Point Net [Qi et al. 2017]).

In other words, the user is expected to decide which models, PyUUL representation and general approach must be used, depending on their requirements and goals.

The strength of PyUUL is that it offers the possibility to rapidly test many cutting edge NN architectures with very little effort. This means that if a bioinformatician finds a new architecture that uses, for instance, point clouds as input in order to perform a prediction and he/she thinks it could be used in a task he is working on, he can directly apply it to biological structures, allowing technology transfer between pure ML or computer vision research and structural bioinformatics.

This might be game changing, because there are currently several competitions about point cloud predictions going on, with several researchers submitting hundreds of new architectures, making it one of the most active fields in computer vision, as described in Bello et al. 2020, even if still partially immature.

We summarized these concepts and explained them better in the paper.

We also added a section in which we use point clouds to address the problem that you suggested in remark number 4 of “NN-based GTP Docking”.

2. Figure 1C, I would recommend reporting the AUC values of the ROC curves.

We moved figure C in the supplementary since we now think it might be misleading. There, we also added the AUC values. We would like to point out that we added the AUC to show that the NN is learning general rules, not to compare it with other docking methods. The NN is still very simple and in order to build an actual docking algorithm it most probably requires the coupling with biophysical approaches such as force fields or energy calculation, which is beyond the scope of this paper. We added this in section S4.2.

3. The alpha helix recognition section under the Neural network example is vague. It is unclear how the model was trained, what training datasets were used, how the model was validated and what predictions were made. I ran the python package and it appears that the section in the main text was a tutorial from the package and therefore I think the scope is somewhat limited for the main text. This is not a complete study and I would suggest expanding on this to showcase the full fold recognition prediction using PyUUL.

This is a toy example to show that PyUUL data structures can be directly used in a 3D-based neural network and that the gradient flows through the volumetric representation. The goal is therefore to show that the parameters of the network are actually trained and that the network is therefore able to overfit. This is an illustrative example on how PyUUL data structures work and it is performed on a single protein. We would like to stress the fact this is just an example: there are better ways to find the alpha helix in protein structures, for instance looking at the Ramachandran phi-psi plot. We wanted to show that the network parameters are updated correctly and provide an easy to use tutorial for the library.

We clarified the explanation in the text.

4. Siamese network for supervised clustering with metric learning section has more description of the models but lacks results.

We added a discussion in supplementary section S4.1. However, in this section we wanted to show that you can use peculiar algorithms that are used in computer vision. We used this example because we think it is interesting that you can easily obtain a “protein signature” similar to molecular fingerprints used for small molecules with ML, just using algorithms that are commonly used in computer vision. This part has therefore the goal to suggest to the reader the type of algorithms that can be used with the volumetric representation of PyUUL is very broad, and they are not limited to the one classically used in bioinformatics.

5.NN-based GTP docking is the section with better coverage.

We added an extra discussion in supplementary material (section S4.2) in which we discuss the network.

The GTP docking has more discussion because this is the part we were most interested in highlighting, since we think it has potential application in end-to-end neural networks (and therefore in the tools that will evolve from AlphaFold, see discussion).

In this part we:

- a) Train a network to recognize the right pose of the GTP with respect to decoys (rotation/translation of the ligand). The training of the network is done (as usual) calculating the gradient of the loss function (binary crossentropy of the aforementioned classification problem) with respect to the **network parameters**. The network parameters are therefore updated at every epoch.
- b) After the training phase is complete, we freeze the network parameters. Then, we take a ligand in the wrong (rotated and translated) conformation, and we calculate the gradient of the network output with respect to a **transformation matrix** (translation/rotation). We therefore change the derivation variable and we obtain a gradient descent optimizer that works on atomic coordinates with minimal coding (the same optimizers used to train a neural network can be used to optimize coordinates). The coordinates of the ligand change with every optimization epoch

This would not have been possible if PyUUL output would not have been completely differentiable, cause the gradient would have stopped otherwise. This is the strength of PyUUL and makes it different from the ad hoc volumetric representations cited on the main paper. Moreover, a completely differentiable volumetric representation is the only one applicable to sequence-to-structure end-to-end neural networks, such as AlphaFold (see discussion).

The models built for each of the three case studies should be more rigorously defined, validated and the analysis/prediction result should all be reported.

The architecture of every network is presented in the methods section and in supplementary material section S3. As for the validation, we added a supplementary section S4 in which we show the results and the problem we had with the development of the network.

An in depth validation and discussion of the networks is beyond the purpose of this paper and it is voluntarily kept short.

6. I was able to run the python package completely without significant issues.

Please let us know if you had (even minor) issues. We are using this library for other projects right now and every usage report is very valuable for us.

NN-based GTP Docking:

1. To improve superiority, perhaps can compare the pose scoring performance with 1 or 2 existing pose scoring functions in current docking programs like DOCK, AutoDOCK etc.

We do not claim superiority of our network respect to the state of the art docking methods. A comparison with docking algorithms would be 1) misleading, since the goal of the paper is to provide a method to convert biological structures in differentiable tensor representations and 2) unfair with general docking methods, since our network is specialized on GTP.

For what concerns the use of NNs in docking, 3D-based neural networks on protein structures have already shown their effectiveness in other works, such as in DOcking [Wang et al. (2020)] in which they used a 3D-convolutional NN on a voxelized pocket, EGCN [Yue et al. (2020)] in which the author use graph convolutional NN (available by applying the pytorch-geometric library to PyUUL outputs), and KDEEP [Jiménez et al. (2018),] in which the authors use 3D-convolutional NN for large scale ligands screening.

The scope of PyUUL is to promote the use of NNs to solve structural bioinformatics problems, not to present a specific tool addressing one of these tasks.

2. The pose prediction result seems good but is done for 1 GTP-bound complex only. To prove generalizability, I would recommend evaluating additional complexes to ensure that the approach will work on any protein-ligand complexes.

The validation is performed on 515 GTP protein complexes. In the video we show a single example, but in the results section we summarize the performances and in figure S2C we show the distribution of the RMSD between predicted and original pose of GTP (discussed in Supplementary Section S4.2).

The goal of the paper is not to build a neural network that can dock small molecules. However, as mentioned above, other authors showed that similar NN approaches can be applied to docking.

Our purpose is to show that the library can propagate the gradient when differentiating respect to the atom coordinates and that it correctly converges to a solution. This would open brand new opportunities to perform end-to-end ML-directed optimization of protein structures, or docking. Fully investigating these possibilities are nevertheless beyond the scope of this paper.

The limited proof-of-concept results cannot be considered a validation, since, as you suggest, a validation should be performed on a large dataset of heterogeneous small molecules. We

discussed possible options for neural networks in docking in section S4.2, in which we also suggest that the pairing of the neural network with a biophysical forcefield might help the network to generalize information.

3. How do the performances differ with respect to protein subtypes and structures? Similarly, how are the 515 protein structures split between training and validation?

The test proposed has been performed with a 5 folds stratified (30% sequence identity) cross validation. When the network converges to a solution, it usually converges to a good solution. We did not observe any bias based on splitting.

However, we saw an influence of the starting position of the GTP in the convergence. This is due to the fact that, if the starting position of the ligand is too far from the optimal position, the gradient of its coordinates goes to zero (for pure numerical reasons). This makes the optimizer unable to move the ligand, since the gradient acting on it is zero. In order to limit this effect, a commonly used technique is to randomize the initial position of the object to optimize. We therefore optimized 10 different random rotations/translations of the input and we kept as final result the one which converged better at the end of the training (meaning that the network assigned it higher confidence). We pointed this out in the paper.

4. Low (<30%) sequence identity does not guarantee low binding sites similarity. I would more rigorously annotate the binding site in the dataset in terms of their local 3D structural similarity.

We selected a 30% of sequence identity threshold to show that the network was learning the actual shape of the binding pocket, and not just the presence of specific amino acid side-chains. The goal was to show, therefore, that the gradient was calculated correctly, and not to perform a validation. However, we understand talking about thresholds might be confusing. We therefore added a sentence to explain the concept.

One interesting application that the authors can potentially explore is to use the PyUUL to compare binding pockets and then correlate that to the RMSD values to the original pocket. Using other published binding site comparison algorithms is also a possibility.

We followed your suggestion: we applied the concept of protein signature we used for supervised clustering, but using point clouds. We use this algorithm, which basically performs an information compression (similar to what is done in [Orlando et al. (2019)] or (in a non machine learning fashion) with molecular fingerprints), to give a “signature” to each binding pocket. We train the neural network to optimize the signature in a way to assign similar signatures to similar binding proteins. Again, this is just an example of usage. To develop a proper “binding pocket classification tool” an ad-hoc architecture needs to be developed, but indeed PyUUL can be used for structure-to-tensor conversion. The proof of concept network is presented in sections “Proof of concept 3: Protein signature encoding with point clouds”,

“FoldNet for GTP binding pocket signature encoding” in the methods section and supplementary material S4.3.

5. The PyUUL framework applied on GTP-protein complex seems like an interesting application as it opens door to potential applications of NN for drug design and optimization. My main concern is the potential applicability of the framework for novel ligands without a known co-crystal complex. I think that very limited training data size for most of the ligands could potentially diminish the performance and the applicability of the framework.

This is indeed a problem shared by every computational method: the less information we have about a specific complex/protein/ligand the lower the performances will most likely be. Pure machine learning methods are, however, particularly affected by the scarcity of data. As a personal opinion, I think a pure computational approach in a general docking algorithm would have difficulties in converging to a meaningful and general solution due to the excessive degrees of freedom. What I would do is to couple a machine learning approach, conceptually similar to the one we give as example but with an ad hoc architecture, with a biophysical model such as a forcefield, thus constraining further the space of the solutions. The presence of biophysical constraint would reduce the solution space in which the network has to work, consequently reducing the required number of learnable parameters (and therefore the amount of needed training data).

6. Similarly, could the approach be applied to GTP analogs docked to the same site?

I definitely think it can be done with an actual docking neural network (as mentioned above, most probably coupled with a forcefield). The other works mentioned above show extremely good results for 3D-convolutional NN and graph-convolutional NN applied to docking, even if they are often difficult to use (no general and easy to use way to deal with protein structures, one of the things we want to address with this library).

The example network we provide voluntarily addresses an oversimplified problem and most probably it would not be sufficient to address such a more complicated problem.

or different GTP conformations considering ligand flexibility?

When performing the conformation optimization, from the computational point of view, you calculate the gradient of the network output with respect to an array linked in some way to the ligand. In our case, our array was a rotation and a translation. There is, however, nothing different in adding additional values to that array that define, for example, the dihedral angles of the GTP structure. Another thing that could be in principle done (assuming the fact that a fine tuned ad-hoc network architecture able to learn the docking problem has been developed) is to include as parameters to optimize the bond length, allowing the network to stretch or contract the bonds. This, of course, would need to be coupled with some biophysical rule in order to avoid the network generating physically impossible structures.

7. “For every protein, we rotate the GTP at a random angle, and we train a 3D convolutional NN to recognize the “wrong” conformations from the original ones.” Were there only one or multiple

sampled conformation? It is still possible that a random rotation could still generate a “close-to-correct” conformation if the rotation is small unless an RMSD cutoff was used.

We do not use any RMSD cutoff. We decided not to use a RMSD cutoff in the training of the network because 1) The probability of having a rotation and translation very close to 0 is very low 2) even if it was happening, it would not affect the validation, since the validation is performed on the optimization part, and there we randomly rotate and translate the input in order to explore in a more efficient way the solution space.

8. “we ask the NN to put it back to the optimal place, effectively performing an optimization of the GTP-protein binding.” I don’t think this is an accurate description. The NN does not predict transformation but only predicts the score for a particular transformation.

We agree this part is not accurate. The network predicts the score and the optimizer is the one in charge of moving the ligand. We changed this in the text.

Minor comments:

There are multiple typos found throughout the text:

1. Figures 1C 3 and 4 shows... -> show

This has been corrected.

2. More information are available in Supplementary Figure S1 -> is available

This has been corrected.

3. The NN consists in 3 layers of 3D convolutional modules -> consists of

This has been corrected.

4. The NN is thus composed by a spatial transformer layer -> composed of

This has been corrected.

5. 313 out of 515 of the optimized complexes had a RMSD lower than 2 with respect to the original crystal structure -> an RMSD lower..

This has been corrected.

6. but its minimization would have the same effect of ΔG protein optimization. -> as ΔG protein optimization.

This has been corrected.

7. An important problem in structural biology is the optimization of a protein or a drug in function of some predefined objective function, such as a force field. -> using some predefined...

This has been corrected.

8. This protein-ligand optimization can be done simply calculating the gradient of the scoring function respect to the GTP coordinates. -> with respect to the GTP coordinates.

This has been corrected.

9. The input 3D structure of a protein is represented in a voxel grid, where each channel show the occupancy of nitrogen, carbon, and oxygen atoms. -> shows

This has been corrected.

Reviewer #2 (Expertise: Chemoinformatics, Computer Science):

The authors present a python library named PyUUL to allow biological structures in PDB format to be converted to a format suitable for deep learning libraries, within PyTorch. They describe the usage of this library on a few sample problems.

My understanding from the paper is that the main contribution of the paper is software to convert the PDB representation into one of three representations (voxel-based, point-cloud surface, volumetric point-cloud). If my understanding is correct, I would say that the contribution is modest, but useful.

Our library provides indeed the conversion, but the most important part of the library is that **the volumetric transformation is performed in a differentiable way**, meaning that the gradient calculated by Pytorch can pass through it. This is essential for the development of end-to-end sequence-to-structure neural networks, since in this type of architecture the gradient is supposed to “flow” from the inputs to the loss function without interruptions.

While this article was under revision, AlphaFold paper [Jumper, et al. 2021, *Nature*] was published and it turned out it was a sequence-to-structure end-to-end neural network. We actually developed PyUUL because we needed a standardized library to build a sequence-to-structure end to end network for protein loop design and optimization.

After the stunning achievements of AlphaFold, it is likely that this class of neural networks will get very popular in the scientific community. PyUUL could potentially be used to include a 3D-based intermediate step (with a 3D-conv structure for instance) into the AlphaFold architecture. This would not be possible if PyUUL was just a structure to volume python conversor. We clarified this important aspect in the discussion.

Perhaps more detail about the input PDB format and the transformation algorithms could be included.

The library takes as input the standard files that can be downloaded from PDB (such as .pdb, .ent or (with the new update) .sdf files). Moreover, the network can take a list of coordinates as a tensor and the atom names, so basically it allows the user to build his own parsers for more exotic data formats. This is explained in the library documentation, along with a practical guide.

We also included a parser to handle small molecules in SDF format completely out of the box and updated the documentation accordingly.

Moreover, the library is under active development (we are currently using it to develop an end to end sequence-to-structure NN for loop modeling) and we will include new formats in accordance with the suggestions of the users.

It is not clear how comprehensive this capability is. Perhaps the authors could provide in a table all of the deep learning techniques that can be utilized as a result of their work.

This is a general library, every pytorch architecture that takes as input voxels or cloud points is suitable. We addressed this in the main text and we provided some suggestion of where to find architectures (there are authors that provide collections of architectures) and supportive libraries, such as torch-points3d and pytorch-geometrics, that provide tools (developed for computer vision and robotics) to deal with peculiar networks layers (i.e. convolutional graph neural networks). Ideally, PyUUL will allow researchers to use existing NN models in a plug-n-play fashion.

I am also unclear whether this package has been used by other researchers besides the authors. If the authors can show this, that would be a big plus.

Currently, only the authors and some of their collaborators are using this library because this is a brand new tool that has never been published before and therefore nobody had access to it yet. We therefore cannot already have a user base before publishing the paper. However, we are indeed using it to develop a sequence to structure end-to-end applied to loop modeling.

Finally, the paper is readable, but the clarity could be improved, perhaps with the help of a professional proofreader.

We improved the text quality, also in accordance with what was pointed out by reviewer 1 and 3.

Reviewer #3 (Expertise: computational chemistry):

Orlando and co-authors present a library that allows to process 3D information of protein and small molecules and to make it “machine-readable” and suitable for modern end-to-end deep learning algorithms. This is very important given the recent progress made by AI in the field of protein structure prediction, synthesis prediction and drug design. The open-access character of the platform and the promised flexibility make it promising for accelerating the impact of AI in the life sciences, in a multitude of structure-based tasks. The paper is well written and very clear in several technical explanations, which make it accessible to a broad readership. I only have a few comments to further improve its readability and accessibility to less expert readers.

Major comments

- Gradient-based protein structure optimization with PyUUL: it is not clear what authors mean with “wrong” conformations. Please consider improving the description of the goals and the procedure.

This is indeed a very vague term. We meant the conformations in which the GTP has been rotated with respect to the conformation observed in the crystal structure. We changed the text accordingly.

- Overall structure: Some elements of the presented platform are just mentioned in the general introduction and then left in the methods (e.g., surface point cloud). This might be misleading for readers and “hide” important features. I suggest putting the more “generic” description of such representations in the main text, while still leaving the technical details in the methods section.

We added a new section in the results in which we show an application of point clouds. We therefore presented an architecture that can be used with both volumetric and surface cloud points. Moreover, we added a paragraph about the application of PyUUL to non protein data, we added a parser for SDF files and we added a guide for small molecules in the documentation.

- The authors mention the applicability of their pipeline to small molecules as well, but this part has not been really explained in the manuscript. There is also a lack of references to the recent advances in the field of AI for small molecules. The authors could consider expanding upon such statements to improve the paper clarity and provide a reasonable overview of the real possibilities of the proposed library.

The library is meant to be as general as possible, taking a list of coordinates and atom names as input (that's the reason why the PDB parser that reads the PDB structures is kept separated with respect to the rest of the library). This means that, if the radius of each atom and, eventually for voxel representation, the channel is present in the library (see supplementary section S3) PyUUL will handle them. This includes, of course, also small molecules. We focused more on proteins only because it is our main research field.

We already included in the library the data about the most common atoms that can be found in PDBs. These atoms can be handled completely out of the box with PyUUL parser (exactly the same as proteins), but the user can handle more exotic atoms bypassing the parser and adding them manually as parameters of the main pyuul python object.

We added a section in the documentation on how to handle these cases. However, we updated the library and we included an out-of-the-box parser for SDF files.

We added a section in the main text and in the documentation in order to guide the user in dealing with small molecules.

Minor comments

- Briefly explain the concepts behind Siamese networks at the first mention for less expert readers

A small description has been included.

- Briefly explain the concepts behind spatial transformers at the first mention

A small description has been included.

- Why is the learned latent space 5D? Consider explaining it at the first occurrence

We simply had severe convergence issues with a lower number of dimensions. This is due to the fact the network has less dimensions to encode the structural information. We pointed this out in the text.

- Explain what a PCA is

A small description has been included.

- Define all acronyms at the first occurrence (e.g., GTP, NN, PCA, pyUUL [if any])

This has been corrected.

PyUUL refers to the friulian (a Rhaetian language mainly spoken in the mountainous region of the north-east of Italy) word *paiûl* and it means pot. We chose this name because 1) it starts with *py*, and it is written in python and 2) it is built over pytorch, which uses a flame as a symbol. Being a pot, a *paiûl* is meant to be put on a flame (PyUUL is indeed based on pytorch) to cook food (data in this case).

- In addition to AlphaFold, also consider citing this paper as well: Baek et al. Accurate prediction of protein structures and interactions using a three-track neural network. Science. 2021 Jul 15.

Done.

Reviewers' Comments:

Reviewer #1:

Remarks to the Author:

The revised manuscript has been improved with an additional application for pocket signature comparison. Below I have some additional minor comments:

1. Figure 2 is not referenced anywhere in the text. I think 2B and C should be switched based on their order in the text. There are also redundant labels such as videos 22, 23, please remove them.
2. "First of all, for each GTP binding protein under scrutiny we calculate the surface point cloud representation using PyUUL." -> calculated.
3. The network takes as input a cloud point and it generate a signature of a fixed size. -> generates

There are multiple places in the text with typos or grammatical issues, please carefully revise.

Reviewer #3:

None

REVIEWERS' COMMENTS

Reviewer #1 (Remarks to the Author):

The revised manuscript has been improved with an additional application for pocket signature comparison. Below I have some additional minor comments:

1. Figure 2 is not referenced anywhere in the text. I think 2B and C should be switched based on their order in the text. There are also redundant labels such as videos 22, 23, please remove them.

Figure 2 has been reworked in order to remove embedded videos. We removed all the redundant labels and figure 2 is now referenced in the main text. We also added references to the supplementary movies 1,2 and 3

2. "First of all, for each GTP binding protein under scrutiny we calculate the surface point cloud representation using PyUUL." -> calculated.

Thanks. We corrected the typo.

3. The network takes as input a cloud point and it generate a signature of a fixed size. -> generates

Thanks. We corrected the typo.

There are multiple places in the text with typos or grammatical issues, please carefully revise.

We revised the full manuscript.